# Kinematic-Based Multi-Objective Design Optimization of a Grapevine Pruning Robotic Manipulator

**Faezeh Molaei [1] and Shirin Ghatrehsamani [2,\*]**

[1] Department of Mechanical Engineering, Isfahan University of Technology, Isfahan 84156-83111, Iran; faezehmolaei@alumni.iut.ac.ir

[2] Agricultural Education and Technology Department, College of Agriculture, Montana State University, Bozeman, MT 59717, USA

\* Correspondence: sh.samani@montana.edu

**Abstract:** Annual cane pruning of grape vineyards is a time-consuming and labor-intensive job, but no mechanized or automatic way has been developed to do it yet. Robotic pruning can be a perfect alternative to human labor. This article proposes a systematic seven-stage procedure to design a kinematically optimized manipulator, named 'Prubot', to manage vineyards' cane pruning. The manipulator structure was chosen, resulting in a 7R (Revolute) manipulator with a spherical shoulder and wrist. To obtain the design constraints, the manipulator task space was modeled. The robot's second and third link lengths were determined by optimizing the global translational version of the measure of manipulability and the measure of isotropy of the manipulator arm section. Finally, simulations confirmed the appropriateness of the manipulator workspace. Furthermore, sampling-based path planning simulations were carried out to evaluate the manipulator's kinematic performance. Results illustrated the impressive kinematic performance of the robot in terms of path planning success rate ($\cong$ 100%). The simulations also suggest that among the eight single-query sampling-based path planning algorithms used in the simulations, Lazy RRT and KPIECE are the best ($\leq 5$ s & $\sim$100%) and worst ($\geq 5$ s & $\leq 25\%$) path planning algorithms for such a robot in terms of computation time and success rate, respectively. The procedure proposed in this paper offers a foundation for the kinematic and task-based design of a cane pruning manipulator. It could be promisingly used for designing similar agricultural manipulators.

**Keywords:** multi-objective optimization; kinematic design; agricultural robot; manipulator; grapevine pruning; manipulability; design procedure; sampling-based path planning

## 1. Introduction

Due to population growth and labor shortage, introducing new technologies into agriculture is inevitable. Grapevine winter pruning is a labor-intensive and time-consuming job executed annually for both major methods of pruning: spar pruning and cane pruning [1]. Some machines are able to carry out spar pruning partially, but manual labor is needed to complete the task [2]. However, because cane pruning needs specific expertise and the working environment is unstructured, there are no machines able to work on this task. The only solution is to apply robotics.

An integrated pruning robotic system may consist of cameras, manipulators, a computer, sensors, and a cabin platform containing other system parts and straddling the vine row [3,4]. Few studies have been reported on a robotic pruning system [3,5–10], most of which did not focus on the manipulators' mechanical design—which can significantly enhance the manipulator's operational performance—and rather concentrated on machine vision, algorithm design, and path planning issues. Furthermore, a cane pruning robot has never been designed; the main focus of previous studies was on spar pruning [3,9]. The only mechanical design of a pruning robot was reported by Zahid et al. (2020) [7], in



which a three rotational (3R) end-effector was proposed for apple tree pruning. The end-effector was integrated with a three prismatic (3P) manipulator to form a pruning robot. Simulations and field tests evaluated the manipulability performance and workspace of the end-effector and the integrated robot. Nevertheless, the authors did not state the design procedure or the method they came up with this design for the end-effector. In addition, a commercial vineyard robotic system is also designed by Vision Robotics Corporation (VRC) [4], but detailed information is not available.

Although some research has been carried out to develop and introduce proper mechanical design indices [11–15] and methods to globalize them [16] for manipulators, many engineers still design manipulators based on empirical estimations [17], which could lead to a robot's inappropriate operational performance. In order to meet the design needs for a robotic manipulator, proper design indices should be taken into account to be optimized in a single-objective or multi-objective optimization process.

Serial manipulators are suitable for light-duty low-energy tasks. Still, parallel manipulators are better choices due to their ability to withstand heavy loads when it comes to heavy-duty ones. In addition, serial ones can offer a larger workspace and dexterity compared to parallel manipulators [18]. Since a grapevine-pruning robot is a light-duty robot that needs high maneuverability to be capable of working in its unstructured and cluttered task space and must have a large workspace that holds all or a major part of the vine body, it must be a serial manipulator to fulfill the job appropriately.

For light-duty manipulators, kinematic design is a major part of the design process [19]. Thus, the following is a literature review regarding design optimization of the non-agricultural and agricultural serial manipulators, especially those concentrating on kinematic design. Harvesting robots are roughly similar to pruning robots in terms of the task itself and the working environment. In recent decades, researchers have put a lot of effort into harvesting robots, though only a few studies proposed design procedures based on proper mechanical design indices [20–25], and it seems others only designed their manipulators empirically and evaluated the mechanical performance to validate the design [26]. A review of harvesting and pruning robots can be found in [27–29].

Limited studies have been carried out on multi-objective design optimization of non-agricultural manipulators with manipulator design indices as their objective functions [30–32], none of which have designed the robot for a specific job but rather only aimed to enhance the performance indices (i.e., condition number, manipulability, etc.) of a commercially available industrial manipulator. At the same time, it may result in an inappropriate workspace for some specific tasks. Even the index-based design optimization procedures proposed in the abovementioned studies of harvesting manipulators [20–25] and non-agricultural manipulators [30–32] suffer from one or both of these major drawbacks: (i) not globalizing or appropriately globalizing the design indices and (ii) using single-objective optimization methods for solving multi-objective optimization design problems, which leads to missing the design Pareto front of trade-off answers which, otherwise, could provide the designer with the opportunity of choosing from a range of optimized answers. Most agricultural robots, especially tree pruning and tree harvesting robots, should be designed through an optimization task-based approach. This approach decreases the design cost function and results in a more suitable and optimized robot for the job [24]. Nevertheless, no systematic design procedure has been proposed in previous studies. Such a procedure can pave the way for manipulator design for engineers and researchers. There might be a specific design approach in some research [21,22,24] but not a systematic and well-defined procedure.

In this paper, a task-based kinematic design of a grapevine cane pruning manipulator was proposed for the first time. The design was carried out according to a well-defined systematic design procedure which was introduced for the first time in this paper and adds a certain novelty value to the study. First, the manipulator structure which is a 7R (Revolute) serial one was chosen. Then, the lengths of the second and third links were determined by optimizing two well-globalized translational manipulability indices. Other

manipulator parameters were determined according to their task space and duty. Simulations proved that the robot workspace was appropriate, and the robot's kinematic performance was evaluated by sampling-based path planning simulations, which indicated the robot's impressive kinematic performance. The proposed design procedure in this paper offers foundations for task-based and kinematic design of tree pruning manipulators and other similar manipulators, including tree harvesting robots.

## 2. Design Procedure

The following is a systematic procedure for task-based kinematic design optimization which can be used to design any agricultural manipulator. It was utilized to design the serial pruning manipulator in this paper. The prototyping and modeling (i.e., the first and second) stages of this procedure establish the connection between the manipulator and its task.

***I Prototyping*:** The first step of the design procedure is to empirically choose or define a suitable joint arrangement for the manipulator. In other words, the number and type of joints (i.e., prismatic or rotational) and the way they are arranged is defined according to the robot's task and environment. This is the common method used in designing agricultural robots [22,24,25,33]. In this stage, the manipulator link lengths and/or the unknown angles between joints' axes are defined as the design parameters.

***II Modeling*:** This stage is the major part of a task-based manipulator design. The manipulator and its task space must be modeled in order to set the robot position with respect to the tree and derive the design constraints, especially constraints on link lengths. This can be performed by one of the following approaches:

A. Measuring the structure of several sample trees and the vineyard's (or orchard's) environmental properties. The data are used to derive appropriate models. Until now, this method has been the most accurate method of agricultural robot task space modeling offered and was proposed for the first time by Bloch et al. in 2015 [24].

B. The designer can estimate an overall and rough model for tree and task space according to the trellising and pruning methods of the orchards and vineyards that the robot will work in. This is not a precise method, though it can lead to a typical model that is helpful in the manipulator design process.

***III Deriving design optimization objective functions*:** Suitable kinematic design indices [11–15] shall be chosen as the optimization design objective functions. Since a pruning or harvesting manipulator works in a cluttered and unstructured environment, it is essential to consider kinematic manipulability indices. Generally, the more design indices are considered, the more precise and suitable the design is.

***IV Extra design considerations*:** Depending on the design problem, there may be some design considerations, such as manipulator body shape and joint ranges, that remain to decide on. This stage is required to complete some of the following stages, including stages V and VII. Hence, depending on the manipulator in hand, this stage may be carried out at any design phase, and it does not necessarily happen at stage IV, as stated here.

***V Globalizing the objective functions:*** In contrast to industrial manipulators, agricultural manipulators work in a cluttered and unstructured environment, and their workspace is not restricted to some specific points. Therefore, it is crucial to globalize the local design objective functions for achieving a globally optimized manipulator [16].

***VI Optimization problem solving*:** So far, the objective functions and constraints of the design problem are defined. Now a suitable optimization algorithm must be chosen to solve the problem. In the case of a design problem with more than one objective function, it is vital to use multi-objective optimization methods to avoid missing the Pareto front of trade-off answers, which offers the opportunity to choose among a range of optimized solutions [34]. At the end of this stage, the design answer is obtained.

***VII Evaluations*:** After the design of a manipulator is completed in terms of all aspects of kinematics, dynamics, control, links strength, etc., it is essential to evaluate the overall performance of the robot in a field test in the real world. However, when it comes

to only kinematic design, as proposed in this study, simulations can be a good tool for evaluating the manipulator's kinematic performance, including workspace and manipulability. Kinematic simulations have been used in agricultural manipulator assessments by other researchers [8,22,24,35]. They let the designer correct the kinematic design where necessary and then go forward to the following design aspects: dynamic, control, etc. Moreover, in problems with two or three objective functions, comparing the obtained Pareto front with the Phenotype space (i.e., the feasible design space or a space where the coordinate axes are the design objective functions), is beneficial to validate the optimization step visually. At the same time, it is not enough to evaluate the manipulator kinematic performance because of the simplifications and approximations that might have been applied during different steps of the design procedure. Hence, a kinematic simulation is also required to validate the kinematic performance of the designed manipulator.

To the best knowledge of the authors, there is no method to find the optimized joint arrangements for a manipulator. Hence, the researchers empirically choose or define an arrangement based on the task. To find the best joint arrangements, one must design several manipulators with different joint arrangements according to the above procedure and finally choose the manipulator with the best magnitudes of objective functions [22,36,37].

## 3. Prototyping

In grapevine cane pruning, most of the cut-points are near the trunk body and vine head (i.e., the upper part of the trunk). In addition, many canes grow out of or near the vine head, which makes it a dense area for the pruning manipulator. A grapevine cane pruning manipulator is equipped with a sharp shear-like end-effector. Therefore, it must be highly manipulative to avoid collisions with the vine trunk, head, and canes. Otherwise, the collision can hurt the dormant buds, hence impacting vine growing and fruiting negatively. Therefore, such a robot should be a kinematically redundant manipulator. A 7-Degrees of Freedom (DoF) manipulator, with one DoF of redundancy, could be a good choice, since more degrees of redundancy increase the trajectory planning and design computational costs extremely. In addition, this manipulator needs a vast workspace to reach all the cut-points (this will be clarified in Section 4). In research by Hollerbach [38], the researcher tried to find an optimum structure for a 7-DoF redundant manipulator by taking four major criteria into consideration including (1) elimination of internal singularities by increasing the manipulator manipulability, (2) optimization of the workspace by offering a structure with a vast workspace, (3) easily solvable kinematic equations for decreasing the computational costs, and (4) simple mechanical construction. Eventually, the design led to a manipulator structure with a 3-DoF spherical Roll-Pitch-Roll type wrist and shoulder and a revolute 1-DoF elbow, as shown in Figure 1. All the manipulator's joints are revolute because revolute joints offer both translational and rotational velocities in the end-effector as well as a larger workspace, while prismatic joints only offer translational velocities in the end-effector and a smaller workspace [38]. Taking all these into consideration, we chose this structure as the prototype for the cane pruning manipulator, so the link lengths are the design parameters.

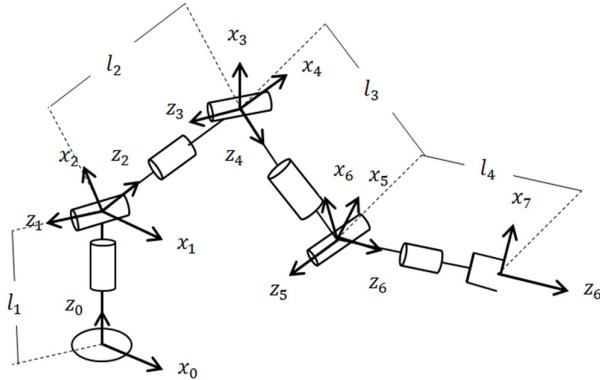

**Figure 1.** Structure of the manipulator prototype.

The Denavit–Hartenberg (D-H) parameters of the prototype in Figure 1 are listed in Table 1, from which the manipulator's Jacobian matrix is derived using Maple 18 software and is shown in Equation (1). The matrix is provided in the Supplementary Materials. In Table 1, $\theta_i$ is the angle from $x_{i-1}$ to $x_i$ about $z_{i-1}$, $\alpha_i$ is the angle from $z_{i-1}$ to $z_i$ about $x_i$, $d_i$ is the distance from $x_{i-1}$ to $x_i$ along $z_{i-1}$, and finally $a_i$ is the distance from $z_i$ to $z_{i-1}$ along $x_i$ [39].

**Table 1.** The Denavit–Hartenberg parameters of the prototype in Figure 1.

| $\theta_i$ | $\alpha_i$ | $d_i$ | $a_i$ | Joint |
|---|---|---|---|---|
| $\theta_1$ | $\dfrac{\pi}{2}$ | $l_1$ | 0 | 1 |
| $\theta_2$ | $\dfrac{\pi}{2}$ | 0 | 0 | 2 |
| $\theta_3$ | $\dfrac{-\pi}{2}$ | $l_2$ | 0 | 3 |
| $\theta_4$ | $\dfrac{\pi}{2}$ | 0 | 0 | 4 |
| $\theta_5$ | $\dfrac{-\pi}{2}$ | $l_3$ | 0 | 5 |
| $\theta_6$ | $\dfrac{\pi}{2}$ | 0 | 0 | 6 |
| $\theta_7$ | 0 | $l_4$ | 0 | 7 |

$$J = \begin{bmatrix} [J_T]_{3\times 7} \\ [J_R]_{3\times 7} \end{bmatrix} \tag{1}$$

where $J_T$ and $J_R$ are the translational and rotational Jacobian sub-matrices of the manipulator in Figure 1, respectively and the Jacobian properties are as follows:

1. None of the elements of $J$ are dependent on $l_1$, thus $l_1$ cannot be determined through optimizing performance indices derived from $J$;

2. Every $l_4$ in the elements of $J$ is multiplied by at least three sines and/or cosines, hence for simplicity, terms containing $l_4$ could be ignored;

3. None of the elements of $J_R$ are dependent on manipulator link lengths, thus in order to determine the robot's link lengths, there is no need for taking this sub-matrix into account.

Hence, the manipulator design parameters boil down to the lengths of the second and third links, i.e., $l_2$ and $l_3$.

## 4. Modeling

Despite the fact that the first approach of modeling introduced in Section 2 is more accurate, in this study the second approach is used because the first approach requires a

specific measurement device which was designed by Bloch et al. in 2015 [24] and the authors did not have access to this device. Therefore, the modeling is carried out based on the information in scientific texts and the authors' previous experiments in vineyards. In Figures 2 and 3, a simple task space model of a grapevine winter cane pruning robot and the pruning cut points are illustrated according to the vine anatomy, trellising system, and pruning principles stated in [1,40–42]. In Figure 2 inline post, trellis wires, vine trunk, vine head, two-year-old canes, canes, the ground, robotic system, manipulator shoulder, cane cut-points, head cut-points, and trunk cut-points are illustrated. According to the pruning principles [1], after pruning, the vine will only have two canes and two short spares attached to the vine head. Then, the two canes must be laid on Wire1 horizontally in a way that they become the following year's two-year canes.

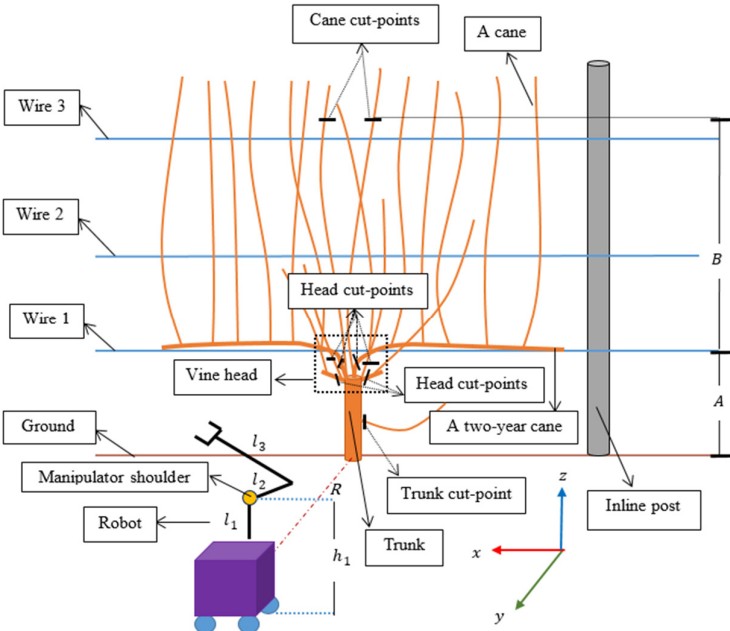

**Figure 2.** Robot winter cane pruning task space.

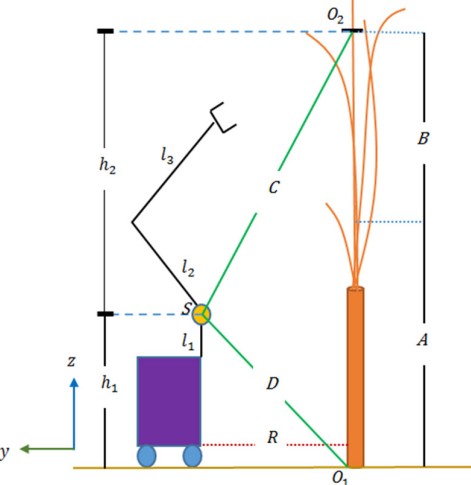

**Figure 3.** Robot winter cane pruning task space in the yz plane.

As illustrated in Figure 2, in the vine trellising system each vine is approximately symmetrical with respect to the yz plane that passes through the middle of the vine trunk, hence the manipulator's shoulder is located exactly in front of the trunk and has a distance of $R$ from the trunk. In Figure 2, $O_1$ shows the point where the trunk meets the ground, $O_2$ is the cane cut-points, $S$ is the manipulator's shoulder, $A$ is the vertical distance between the ground and wire 1, $B$ is the vertical distance between wire 1 and $O_2$, $C$ is the distance between $S$ and the point $O_2$ in the yz plane, $D$ is the distance between $S$ and the point $O_1$ in the yz plane, $h_1$ is the vertical distance between the ground and $S$, and finally $h_2$ is the vertical distance between $S$ and $O_2$.

In a row of the grapevine trellising system, the vines are about 1.83 to 3.36 m (6 to 12 feet) apart depending on the vine variety [40]. In the case of our pruning robot, this distance is assumed to be 2 m, thus the distance $B$ will be half of it (i.e., $B = 1$ m), because $B$ is the length of one cane that remains after pruning and half of the distance between the trunks of two vines in a row [40,42]. In addition, we assume that the condition $A = 1$ m is also satisfied [42].

The manipulator needs to be mounted on a mobile platform. The platform wheels shall have a distance of at least 0.5 m from the crop row to avoid root system damages due to soil compaction [42] so that the distance of the manipulator's shoulder to the crop row (i.e., $R$) is assumed to be 0.8 m.

In order to prune the vine, the manipulator needs to reach all the cut-points easily. This means that the manipulator must reach the two most extreme cut-points, i.e., the cane cut-points and any trunk cut-point which may grow near the ground. Hence, the magnitude of $l_2 + l_3$ must be determined in such a way that the manipulator is capable of reaching points $O_1$ and $O_2$ in Figure 3. Therefore, $l_2 + l_3 = \max(C, D)$ must satisfy. Obviously when $h_1 = h_2$ is satisfied (i.e., $D = C$ is satisfied), then $l_2 + l_3$ is at its minimum and can be determined as in Equations (2) and (3).

$$h_1 = h_2 = \frac{A + B}{2} = \frac{1 + 1}{2} = 1 \tag{2}$$

$$l_2 + l_3 = D = C = \sqrt{R^2 + {h_2}^2} = \sqrt{0.8^2 + 1} = 1.28 \cong 1.3 \tag{3}$$

To make certain of an adequate reachable workspace, $l_2 + l_3$ could be considered a little more than 1.3 m, for example, 1.4 m. Hence, the design constraint could be defined as Equation (4).

$$1.3 \leq l_2 + l_3 \leq 1.4 \tag{4}$$

In addition, the canes with cane cut-points, are chosen in a way that they do not lean forward or backward too much; otherwise, the canes may break while being laid on wire 1. However, these canes may lean towards the right or left (in Figure 2) and it brings about no difficulty, because if the manipulator length is enough to reach the cane cut-points (i.e., point $O_2$) on a vertical cane, then it will easily reach the cane cut-points on the leaned canes too. Therefore, the constraint in Equation (4) is valid for any situation.

## 5. Deriving Design Optimization Objective Functions

The vine head area is a very dense space for the pruning manipulator since lots of spares and canes grow out of the head and most of the cut-points are located in this area. Because the manipulator is equipped with a sharp shears-like end-effector, the pruning robot must be highly manipulative to work in this area without damaging the dormant buds. The measure of manipulability and the condition number of the manipulator's Jacobian matrix are the two major and well-known kinematic manipulability indices, which are widely used in robot design [7,20,23,30–32]. These indices, which both are local criteria, are taken as the design objective functions for the pruning manipulator and are defined as follows:

*1. Measure of Manipulability*: The further the Jacobian determinant is from zero, the further the robot is from singularity. Hence, the Jacobian determinant could be a good measure of the robot's manipulability. Yoshikawa has defined the measure of manipulability as in Equation (5) [11],

$$w = \sqrt{\det \boldsymbol{JJ}^T} \tag{5}$$

where $\boldsymbol{J}$ is the manipulator's Jacobian matrix and is an $m \times n$ matrix. $m$ is the dimension of the Cartesian space $R^m$ (the work space) of the robot and $n$ is the dimension of the robot's configuration space $R^n$.

*2. Condition number of the manipulator's Jacobian matrix*: This index is sometimes called dexterity or manipulability index and is defined as in Equation (6) [12,15,20,43].

$$cond(\boldsymbol{J}) = \frac{\sigma_{min}}{\sigma_{max}} \tag{6}$$

where $cond(\boldsymbol{J})$ is the condition number of the Jacobian matrix, $\sigma_{min}$ and $\sigma_{max}$ are the minimum and maximum singular values of this matrix, respectively. On the other hand, because the elements of the Jacobian matrix, hence the singular values, are functions of link lengths which are unknown and joint values which are local measures, it is impossible to distinguish between the maximum and minimum singular values. Therefore, the condition number cannot be expressed analytically. To compute the condition number, one method is to use numerical methods, though they lead to high computational costs. The other method is to use the measure of isotropy instead of the condition number, which expresses the same concept as the condition number and is defined as in Equations (7)–(10). In contrast to the condition number, this index uses all the singular values and is expressed analytically, hence reducing computational costs extremely [14]. The measure of isotropy is bounded between 0 for singular conditions and 1 for optimum isotropic conditions.

$$M = \sqrt[m]{\det (\boldsymbol{JJ}^T)} = \sqrt[m]{\lambda_1 \lambda_2 \ldots \lambda_m} \tag{7}$$

$$\Psi = \frac{trace(\boldsymbol{JJ}^T)}{m} = \frac{\lambda_1 + \lambda_2 + \cdots + \lambda_m}{m} \tag{8}$$

$$\Delta = \frac{M}{\Psi} \tag{9}$$

$$\sigma_i = \sqrt{\lambda_i} \qquad \forall i = 1,2,\ldots m \tag{10}$$

where $\lambda_i$ is the $i$th eigenvalue of $\boldsymbol{J}^T\boldsymbol{J}$, $M$ is the geometric mean of the eigenvalues of $\boldsymbol{J}^T\boldsymbol{J}$, $\Psi$ is their arithmetic mean, $\Delta$ is the robot's measure of isotropy, and $\sigma_i$ is the $i$th singular value of the matrix $\boldsymbol{J}$.

The measure of manipulability and the measure of isotropy (i.e., the condition number) represent the volume and isotropy of the manipulability ellipsoid, respectively. By optimizing these two indices, we aim to increase the volume of the ellipsoid and transform the ellipsoid into a sphere [11,20].

As the major translational and rotational movements of the end-effector are performed by the arm section and the wrist section of the manipulator, respectively [44], mechanical design indices derived from the translational and rotational Jacobian sub-matrices could be utilized to design the arm and the wrist of the manipulator, respectively, [13,20]. Moreover, by taking advantage of this concept, there is no need for Jacobian unit normalization [45] for our revolute manipulator.

In a study by Gosselin and Angeles in 1991 [16], it was substantiated that, in terms of global conditioning index (i.e., global condition number), Roll-Pitch-Roll is the optimum structure for an open-loop spherical wrist. Furthermore, in Section 3, we have accepted this wrist structure for the manipulator and now we only need to design the arm section of the manipulator. Hence, for determining the design parameters (i.e., $l_2$ and $l_3$), we

need to optimize the translational manipulability indices of the manipulator. This means that the objective functions must be derived from $[J_T]_{3\times7}$. Thus, the design objective functions of the manipulator are the translational measure of manipulability and the translational measure of isotropy. However, $[J_T]_{3\times7}$ is a $3 \times 7$ matrix and causes cumbersome calculations. Hence, in this study, for simplicity, the arm section of the manipulator is considered an independent manipulator as illustrated in Figure 4; its Denavit–Hartenberg parameters are stated in Table 2, and Equation (11) shows its Jacobian sub-matrices.

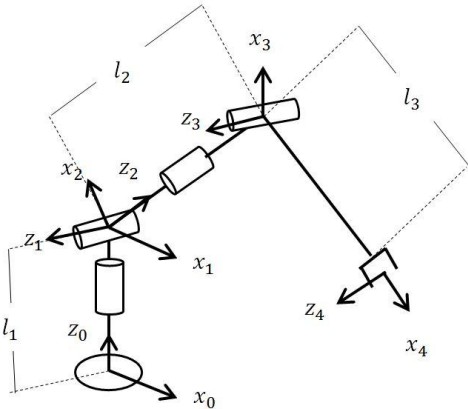

**Figure 4.** Arm section of the prototype.

**Table 2.** The Denavit–Hartenberg parameters of the arm in Figure 4.

| $\theta_i$ | $\alpha_i$ | $d_i$ | $a_i$ | *Joint* |
|---|---|---|---|---|
| $\theta_1$ | $\dfrac{\pi}{2}$ | $l_1$ | 0 | 1 |
| $\theta_2$ | $\dfrac{\pi}{2}$ | 0 | 0 | 2 |
| $\theta_3$ | $\dfrac{-\pi}{2}$ | $l_2$ | 0 | 3 |
| $\theta_4$ | 0 | 0 | $l_3$ | 4 |

$$J' = \begin{bmatrix} [J'_T]_{3\times4} \\ [J'_R]_{3\times4} \end{bmatrix} \tag{11}$$

where $J'$ is the Jacobian matrix of the manipulator in Figure 4 and $J'_T$ and $J'_R$ are its translational and rotational Jacobian sub-matrices. This matrix is calculated in Maple 18 software and is provided in the Supplementary Materials. The Jacobian properties are as follows:

1. $J'$ is not dependent on $l_1$ so this length cannot be determined through design indices derived from $J'$.
2. $J'_R$ is not dependent on the manipulator's link lengths, so to determine the robot's link lengths there is no need for taking this sub-matrix into account.

Because the arm in Figure 4 performs the major part of translational movements of the manipulator in Figure 1 and the indices derived from $J'_T$ can describe the major part of translational movement of the arm in Figure 4, it is demonstrated that the indices derived from $J'_T$ can also describe the major part of the translational movement of the 7-DoF prototype manipulator in Figure 1. In addition, $J'_T$ has three fewer columns compared to $J_T$. Therefore, for the sake of simplicity and declining the computational costs, $J'_T$ could be utilized instead of $J_T$ to compute the translational design indices of the 7-DoF prototype.

## 6. Extra Design Considerations

In addition to determining the manipulator's design parameters, there are some additional criteria that must be taken into consideration in the robot's final design:

1. No holes should exist on the surface of the robot body, especially near its joints, because canes could enter them and damage the robot and/or the vine. Considering the commercially available robots, a good choice is the body design of KUKA LBR iiwa.
2. The manipulator's joints are assumed to have the same ranges as KUKA LBR iiwa's, as simulations indicated that these joint ranges eliminate a vast range of self-collision configurations. The joint ranges are: $Joint_1 = \pm170$, $Joint_2 = \pm120$, $Joint_3 = \pm170$, $Joint_4 = \pm120$, $Joint_5 = \pm170$, $Joint_6 = \pm120$, $Joint_7 = \pm170$. Arranging the joint ranges is vital to globalizing the design indices.
3. The robot links should be as thin as possible in order to make it easier for the manipulator to avoid obstacles. In the present study, the robot link radius is assumed to be 10 cm for modeling and simulation purposes by referring to the properties of commercially available robots.

## 7. Globalizing the Objective Functions

The two indices mentioned in Section 5 are local measures and only can optimally design a robot for some restricted configurations. As it is stated in Section 4, unlike industrial robots, a grapevine winter cane pruning robot works in an unstructured and cluttered environment; thus, designing this robot demands global performance indices. In a study in 1991, Gosselin and Angeles [16] suggested a general method, illustrated in Equations (12)–(14), by which the global version of any local indices could be computed. This calculation results in an index that is independent of robot joint values and is a function of the design parameters.

$$G = \frac{A}{B} \tag{12}$$

$$A = \int_W P \ dW = \int_R P \ |Det\ (\boldsymbol{J})| \ d\theta_n \ldots d\theta_2 \ d\theta_1 \tag{13}$$

$$B = \int_W dW = \int_R |Det(\boldsymbol{J})| \ d\theta_n \ldots d\theta_2 \ d\theta_1 \tag{14}$$

where $P$ is any local performance index and $G$ is its global version, $|Det(\boldsymbol{J})|$ is the absolute determinant of the Jacobian matrix, $W$ is the manipulator's workspace in the Cartesian space, $R$ is the manipulator's configuration space and $B$ is the volume of the robot's workspace.

In this study, the global translational measure of manipulability and the global translational measure of isotropy of the arm in Figure 4 (i.e., the volume and isotropy of the corresponding global translational manipulability ellipsoid) are derived from $\boldsymbol{J}'_T$ and utilized to determine the design parameters, i.e., $l_2$ and $l_3$. The two aforementioned indices have been computed in MATLAB V5 R2016a (MathWorks, Inc., Natick, MA, USA) and Maple 18 and are provided in Supplementary Materials and expressed in Equations (15) and (16),

$$w'_{T_{Global}} = l_2^2 l_3^2 (0.216436574907681 l_2^2 + 0.365860471299765 l_3^2) \tag{15}$$

$$\Delta'_{T_{Global}} = f\left(\frac{l_3}{l_2}\right) \tag{16}$$

where $w'_{T_{Global}}$ and $\Delta'_{T_{Global}}$ are, respectively, the global translational measure of manipulability and the global translational measure of isotropy derived from $\boldsymbol{J}'_T$. The two equations indicate that $w'_{T_{Global}}$ is a function of $(l_2, l_3)$ and $\Delta'_{T_{Global}}$ is a function of $\left(\frac{l_3}{l_2}\right)$.

## 8. Optimization Problem Solving

The design constraints and objective functions were derived in Sections 4 and 7, respectively. In optimization, bounded objective functions are preferable, but the global translational manipulability does not have an upper bound. In addition, the authors prefer minimization, while both the objectives could be optimized through maximization, therefore the objectives should be rewritten to meet these needs. Taking these all into consideration, the design optimization problem can be defined as in Equations (17)–(20). Equation (19) is the design constraint and Equation (20) is the geometric constraint [46],

$$minimize \quad Objective1(l_2, l_3) = e^{-w'_{T_{Global}}} \tag{17}$$

$$minimize \quad Objective2(l_2, l_3) = -\Delta'_{T_{Global}} \tag{18}$$

$$1.3 \le l_2 + l_3 \le 1.4 \tag{19}$$

$$lb = 0.1 \le l_2, l_3 \le 1.3 = ub \tag{20}$$

where all the magnitudes are in meters, and $lb$ and $ub$, respectively, denote the lower bound and upper bound for both $l_2$ and $l_3$.

The design problem is a multi-objective optimization design. Instead of an answer, a multi-objective optimization problem has a set of answers called the Pareto front, which provides the designer with the chance of choosing an optimized answer that best fits the design circumstances. In this study, the Non-Dominated Sorting Genetic Algorithm II (NSGA-II) [47] is used to solve this problem, and the design constraints are also taken into account by applying the method of penalty function. Computations have been carried out in MATLAB V5 R2016a (The MathWorks, Inc., Natick, MA, USA) and provided in the Supplementary Materials.

## 9. Results and Discussion of the Design Optimization

By substituting 1,000,000 feasible random pairs of $(l_2, l_3)$ in the objective functions in Equations (17) and (18) and satisfying Equations (19) and (20), the design feasible space is obtained as in Figure 5. Figure 6 illustrates the section of Figure 5 where the Pareto front is located. Figure 7 indicates the Pareto front achieved by NSGA-II with a population size of 250, 150 iterations, and the crossover and mutation probabilities of 80 percent and 50 percent, respectively. Comparing Figures 6 and 7, it can be seen that the optimization algorithm has found the Pareto front of trade-off answers successfully and the answers in Figure 7 represent the optimized design for the manipulator in terms of the objective functions.

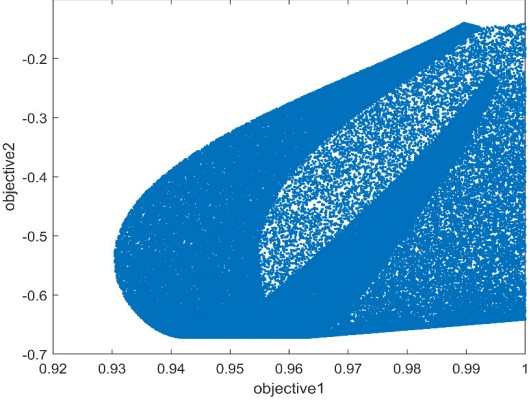

**Figure 5.** The design feasible space of the optimization problem.

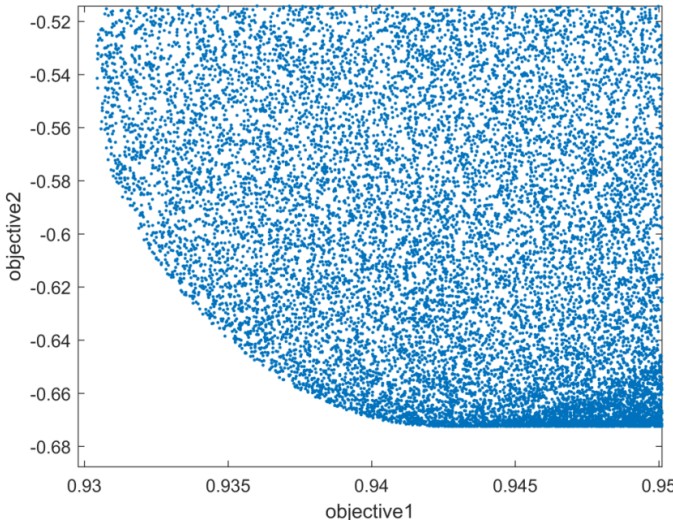

**Figure 6.** The section of Figure 5 where the Pareto front is located.

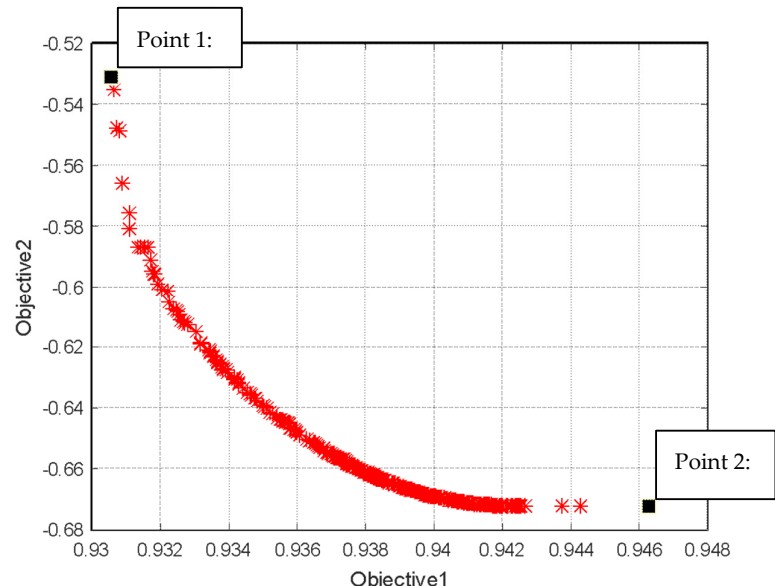

**Figure 7.** The Pareto front of the design optimization problem.

As stated in Equations (15) and (16), objective 1 and objective 2 are functions of $(l_2, l_3)$ and $\left(\frac{l_3}{l_2}\right)$, respectively. If the design optimization was defined as a single objective optimization problem with the volume of the translational manipulability ellipsoid as its only objective function, then the optimized answer would be point 1 in Figure 7 which suggests $l_2 = 0.5553$ m and $l_3 = 0.8442$ m. However, if the objective was the global translational measure of isotropy, the optimized answer would be point 2 of Figure 7, which suggests $l_2 = 0.8289$ m and $l_3 = 0.5547$ m. It demonstrates that although both the objectives represent kinematic translational manipulability, their attitudes toward considering different aspects of the translational manipulability ellipsoid (i.e., the volume and the isotropy of the ellipsoid respectively) lead to a Pareto front of trade-off answers in this study. This Pareto front provides the designer with the chance of choosing an optimized answer that best fits the design circumstances. It proves that considering one of the above objective functions is not enough for an appropriate kinematic design in terms

of kinematic manipulability. However, in some of the previous studies, only one of these indices was taken into consideration and the other one was ignored [23].

Between the two extreme answers of point 1 and point 2 in Figure 7, there are 248 other answers of which 40 answers approximately satisfy $l_2 = l_3 = 0.7$ m and are located in the middle of the Pareto front. In this paper the condition $l_2 = l_3 = 0.7$ m is chosen as the final answer for designing our pruning manipulator which is named 'Prubot' as an abbreviation of 'Pruning robot'. Prubot is illustrated in Figure 8.

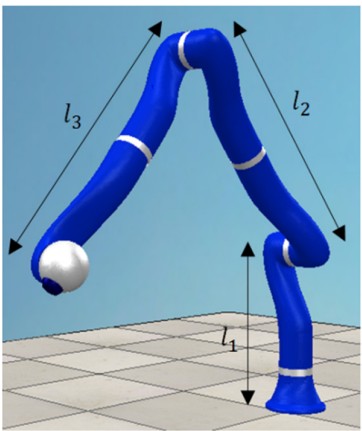

**Figure 8.** Prubot manipulator without end-effector (the body design is based on KUKA LBR iiwa design).

## 10. Simulation Evaluations

The pruning robot and the task space objects (e.g., vines, posts and wires) are modeled in SolidWorks 2013 software and the kinematic simulations are performed using a laptop computer with an Intel Core i7 4500U CPU 1.8–2.4 GH processor and in the environment of CoppeliaSim Edu software version 4.1.0 (rev. 1). The design optimization led to $l_2 = l_3 = 0.7$. At this stage, a 44-cm-long pruning end-effector (i.e., $l_4 = 0.44$ m), designed by the authors in another research, is attached to the manipulator which forms a longer pruning manipulator. Simulations show that the best manipulator performance happens when its base is attached to a wall rather than a floor or a ceiling which is illustrated in Figure 9 in which $l_1 = 0.45$ m must be satisfied to make sure that within the manipulator's second joint's range of movement, the robot elbow does not collide with the wall the robot base is attached to. In fact, the base of the manipulator must be attached to the walls of a platform that straddles the vine row. The platform moves forward as the robot is finished with a vine. The platform is not proposed in this paper but the walls are represented in Figure 9. However, such a platform is provided by other scientists [3,4]. Therefore, the word "wall" does not invoke a static wall, but rather it is an integral part of a platform. The vine and trellising dimensions that are introduced in Section 4 are also used for modeling the robot task space. However, due to attaching an end-effector to the manipulator, it is vital to fine-tune the position of the manipulator's shoulder through simulations. It led to an increase in R (i.e., the distance of Prubot's shoulder from the vine trunk) from 0.8 to 0.96 m.

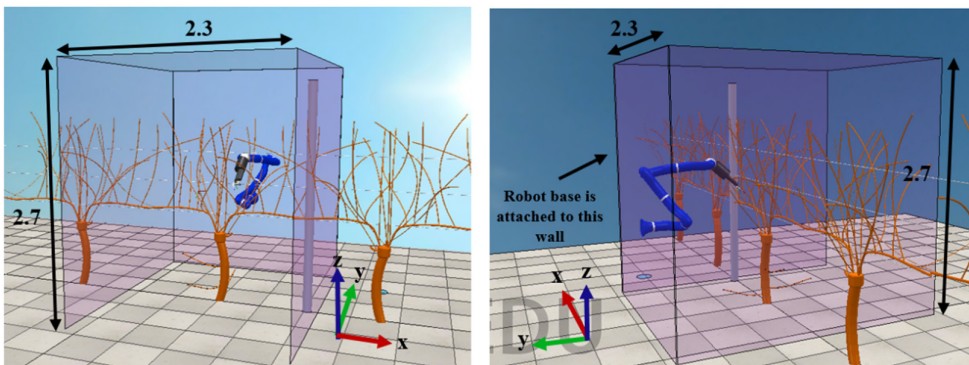

**Figure 9.** Prubot's task space (lengths are in meters).

Prubot's reachable work space is shown in Figure 10 in which 300,000 feasible positions are illustrated by 300,000 random cloud points. The work space contains a vine completely.

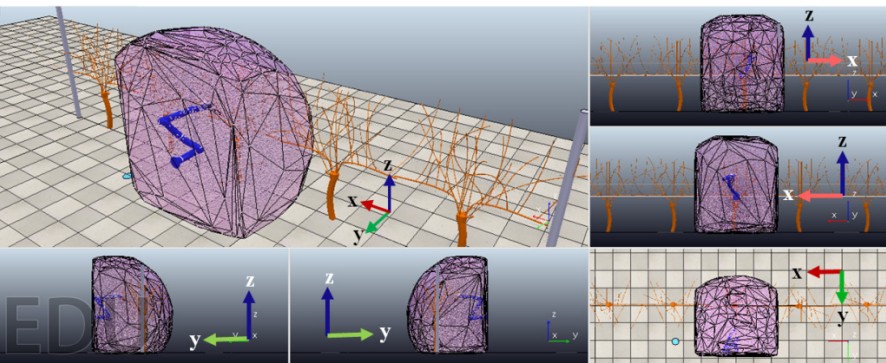

**Figure 10.** Prubot's reachable workspace in different views.

Simulations indicate that to cane prune a vine row from both sides completely, two Prubot manipulators located symmetrically with respect to the crop row are required because the trellising system practically blocks the manipulator's access to the other side of the row. In Figure 11, the dimensions of the task space of a grapevine-pruning robotic system with two Prubot manipulators are illustrated.

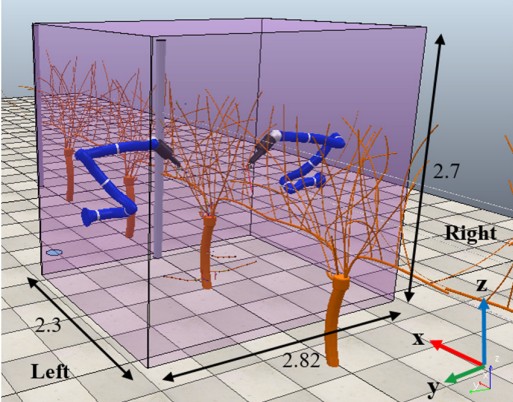

**Figure 11.** The task space of a grapevine-pruning robotic system with Prubot manipulators (lengths are in meters).

A typical vine model is designed according to grapevine anatomy and cane pruning properties [1,41]. This model is used in the simulations. The cut-points of both sides of the vine model are illustrated in Figure 12. Each of the sides is pruned by its corresponding manipulator in Figure 11.

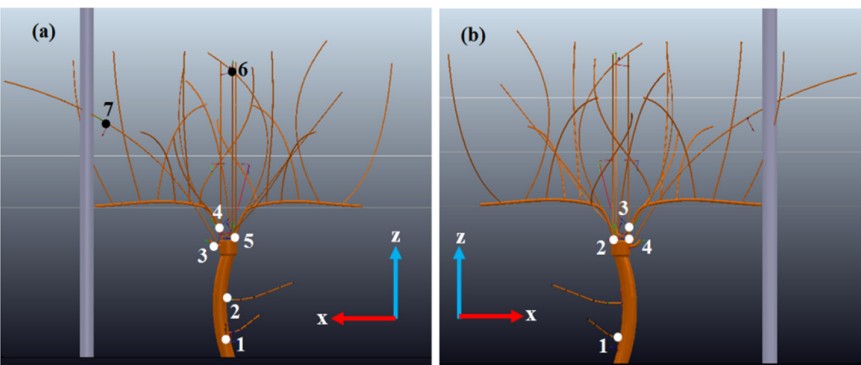

**Figure 12.** The cut-points of (**a**) the left side of the vine model, (**b**) the right side of the vine model.

The kinematic performance of Prubot in pruning the vine model is assessed through path planning simulations in CoppeliaSim software. For this purpose, the script of an example simulation named "motionPlanningDemo1", provided by CoppeliaSim, is used and adjusted for the pruning manipulators. Path planning consists of three phases. In the first phase, the end-effector moves from an initial point to a pre-pruning point with a distance of 10 cm from the target cut-point and an appropriate orientation for cutting the cane, as shown in Figure 13. This phase is planned by a single-query sampling-based path planning algorithm, while collision avoidance (but not self-collision avoidance) is also taken into account. If any self-collision happened during the path, the simulation would be canceled and repeated. In the second phase, which is planned by computing the robot inverse kinematics, the end-effector moves from the pre-pruning point to the target cut-point by traveling along the linear path between the two points. The third phase takes place after the cut and is simply the reverse of the traveled path in the second phase, and finally, the end-effector stops at the pre-pruning point in Figure 13. Now, the robot is ready to plan a path to the next cut-point or the simulation could be stopped.

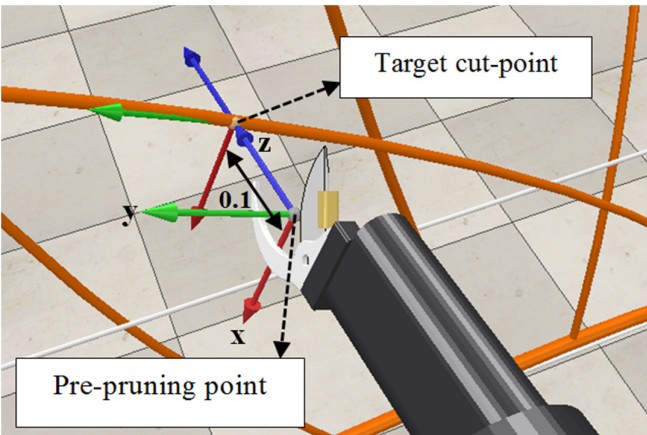

**Figure 13.** The end-effector at the pre-pruning point (Lengths are in meters).

As illustrated in Figures 11 and 12, in the robotic system, the manipulator on the left side of the vine row must prune seven cut-points and the manipulator on the right side must prune four cut-points. Two different simulation tests were carried out in which eight

common single-query sampling-based path planning algorithms are used. Compared to other path planning methods, such as energy field, combinational methods, and multiple query sampling-based (e.g., Road Map algorithm), the properties of the single-query sampling-based path planning algorithms make them the best choice for a pruning robot [8,48]. Eight common single-based sampling-based algorithms, named RRT, LazyRRT, RRTConnect, KPIECE, BKPIECE, LBKPIECE, EST, and SBL, which are also used in previous studies [35,49] are used in this study. The test definitions are as follows:

*1. Test 1 definition*: Test 1 evaluates the success rate of the manipulator's capability of path planning to all its corresponding cut-points consecutively (i.e., the manipulator must prune all its corresponding cut-points, one by one and without interruption) in a way that the initial point of the path planning to a cut-point is the pre-pruning point of the former cut-point. Therefore, most of the end-effector path planning initial points would be in the vine canopy and different for each cut-point's path planning. This is the case for a pruning manipulator in the real world and is the main feature of this test. The test is performed two times per manipulator in the robotic system. The test is also repeated for each of the eight path planning algorithms. To the best knowledge of the authors, it is the first time that such a test has been reported for a pruning manipulator.

*2. Test 2 definition*: In test 2, some simulations are carried out in which the end effector's initial point is out of the vine canopy and the same for all the cut-points. Test 2 is performed once per cut-point in order to determine the path planning computation time and the success rate of the path planning's first phase. The test is also repeated for each of the eight path planning algorithms.

Note that in both test 1 and test 2, a path planning trial is considered to fail if a collision happens or the path planning calculations fail. Although the simulation algorithm script is not an optimized one and it does not lead to a trajectory planning result (the differences between path planning and trajectory are clarified in [48]), it still can be a good tool for evaluating Prubot's kinematic performance and estimating the sampling-based path planning algorithms' performance for Prubot and its task space.

## 11. Results and Discussion of Simulation Evaluations

### 11.1. Workspace

The corresponding simulation to Figure 10 indicates that the work space is so vast that it easily holds the vine entirely and no cut-points are located near or outside of the reachable work space borders, which helps the robot to avoid external singularities. Moreover, the cloud points of end-effector positions are dense in the regions where the cut-points are more likely to exist. Hence, it is demonstrated that Prubot can offer an appropriate workspace for the job.

### 11.2. Test 1

As illustrated in Figure 14, results indicate that in test 1, Prubot is 100% successful in all of the path planning algorithms except in KPIECE and EST, in which success rates are 25% and 75%, respectively.

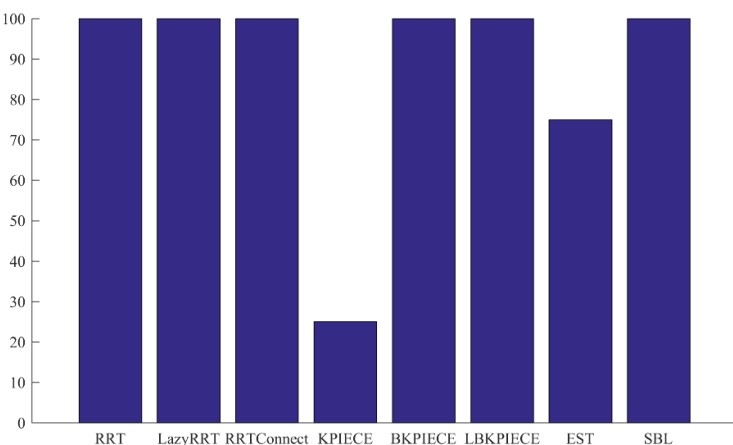

**Figure 14.** Success rate of Prubot manipulator in test 1 for each of the path planning algorithms.

The results of test 1 demonstrate that in the case of utilizing the aforementioned path planning algorithms, apart from KPIECE and EST, Prubot can kinematically and reliably consecutively prune the cut-points in the vine's cluttered environment, where the path planning initial point is in the vine's dense canopy.

### 11.3. Test 2

The success rate of Prubot in test 2 is illustrated in Figure 15. The box plot of the path planning's first phase computation time is also illustrated in Figure 16.

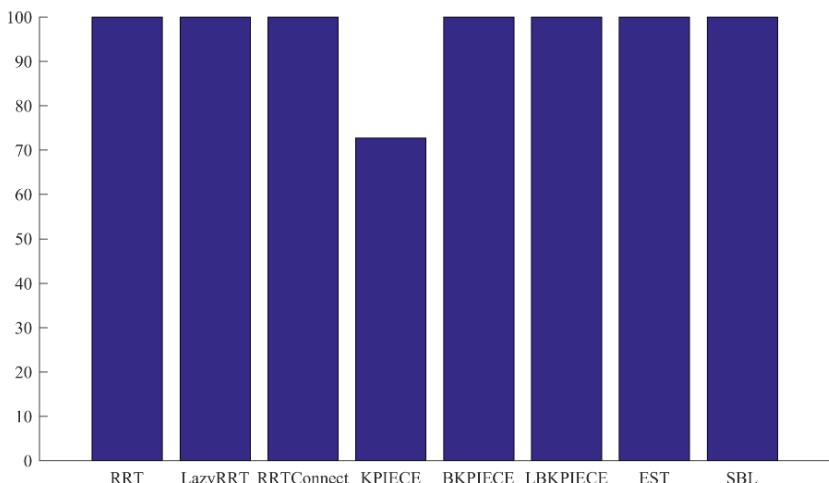

**Figure 15.** Success rate of Prubot manipulator in test 2 for each of the path planning algorithms.

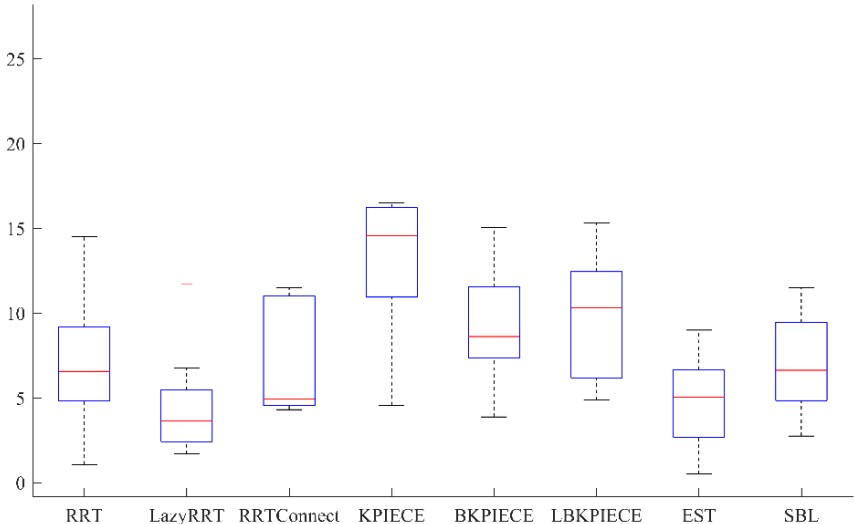

**Figure 16.** Path planning computation time (in seconds) of Prubot in test 2.

The success rate of the algorithms KPIECE and EST are higher in test 2, because in test 1, the initial point of path planning to every cut-point, except the first cut-point, is in the vine's cluttered canopy, but in test 2 the initial point of path planning to every cut-point is out of the vine's canopy.

Figure 16 indicates that Lazy RRT and KPIECE are respectively the best and the worst algorithms in terms of path planning computation time ($\leq 5$ s and $\geq 15$ s) and success rate ($\cong 100\%$ & $\cong 72\%$).

Since in Section 9 the kinematic design optimization of the manipulator's arm performed well and the manipulator's wrist proved to have an optimized structure in terms of GCI [16], the good kinematic performance in the simulations was predictable.

The results of the path planning simulations are highly perfect, because in these simulations, only the kinematic features of the robot are taken into consideration and its dynamics, vibration, control, etc. aspects are ignored, which otherwise could interfere with the perfect results. Nevertheless, in the kinematic phase of design and before the dynamic, control, etc. phases of design, path planning simulations can be a good tool for evaluating the kinematic performance of a manipulator.

## 12. Conclusions and Future Works

In this study, a 7R manipulator, named Prubot, was kinematically designed for cane pruning of grape vineyards through a systematic seven-stage design procedure. First, a suitable joint arrangement was taken as the prototype. Then, task space modeling of the manipulator was carried out to derive the design's constraints. Next, the second and third link lengths were determined by optimizing the global translational form of both the measure of manipulability and the measure of isotropy of the arm section of the manipulator. The optimization process was carried out successfully. Although both the objectives represented a kind of manipulability index, they were directly related to $(l_2, l_3)$ and $(\frac{l_3}{l_2})$, respectively, and led to a Pareto front of trade-off answers from which the final design point (i.e., $l_2 = l_3 = 0.7$ m) was chosen in this study. Finally, the manipulator's work space and kinematic performance were evaluated via simulations which indicated that the design provides a suitable workspace and impressive kinematic performance in terms of success rate. The simulations also suggest that among the eight single-query sampling-based path planning algorithms used in the simulations, Lazy RRT and KPIECE are the best and worst path planning algorithms for such a robot, in terms of both computation

time and success rate. This study can offer a foundation for the task-based kinematic design of a pruning manipulator.

**Supplementary Materials:** The following supporting information can be downloaded at: https://www.mdpi.com/article/10.3390/agriengineering4030040/s1.

**Author Contributions:** Conceptualization, F.M. and S.G.; methodology, F.M.; software, F.M.; validation,F.M. and S.G.; formal analysis, F.M. and S.G.; investigation, F.M.; resources, F.M. and S.G.; data curation, F.M.; writing—original draft preparation, F.M.; writing—review and editing, S.G.; visualization, F.M.; supervision, S.G.; project administration, F.M.; funding acquisition, S.G. All authors have read and agreed to the published version of the manuscript

**Funding:** This research did not receive any specific grant from funding agencies in the public, commercial, or not-for-profit sectors.

**Institutional Review Board Statement:** Not applicable.

**Informed Consent Statement:** Not applicable.

**Data Availability Statement:** Not applicable.

**Acknowledgments:** The authors would like to gratefully acknowledge Mostafa Ghayour (in the Department of Mechanical Engineering, Isfahan University of Technology, Iran) for helping in mechanical and robotic side of the research and Mahdiyeh Gholami (in the College of Agricultural Engineering, Isfahan University of Technology, Iran) for teaching grapevines anatomy, pruning and trellising systems to the authors.

**Conflicts of Interest:** The authors declare no conflict of interest.

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
