# Peer review of "Kinematic-Based Multi-Objective Design Optimization of a Grapevine Pruning Robotic Manipulator"

_agriengineering, doi:10.3390/agriengineering4030040_

Round 1

Reviewer 1 Report

The article aims at the kinematic optimization of a 7R manipulator robot. The simulations are done on a reduced 4R manipulator robot.

Between 227 -249 are presented the Jacobian matrix and the Denavit-Hartenberg (D-H) parameters for a robot with 7 joints. After, between 385 -405 are the Jacobian matrix and the Denavit-Hartenberg (D-H) parameters for a robot with 4 joints that are then modeled further. The model in rows  227 -249 can be excluded and all references will be reduced to a 4R robot.

Also, there are some notations that can create confusion.

In equation 6, parameter σ1 could be renamed σmin, and parameter σm could be renamed σmax

In equations 7-10 which is the meaning of “m”. The notation was used in row 329 as a Jacobian matrix dimension.

In equation 10 what is the meaning of the parameter σi?

In equations 13 and 14 what is the meaning of parameters A and B?

What are the mechanical implications when the real manipulator must work as in the simulation (attached to the wall)?

I think the article can be published with small changes.

Author Response

Dear Reviewer

Sincerest thanks for your response and comments on our manuscript. We have modified the paper in response to the extensive and insightful comments. We have worked on clarification to fully address them. We have stressed the main goal of this paper as the technical presentation of treatment in the abstract and conclusion. 

Comment 1:

Between 227 -249 are presented the Jacobian matrix and the Denavit-Hartenberg (D-H) parameters for a robot with 7 joints. After, between 385 -405 are the Jacobian matrix and the Denavit-Hartenberg (D-H) parameters for a robot with 4 joints that are then modeled further. The model in rows 227 -249 can be excluded and all references will be reduced to a 4R robot.

Response to comment 1:

As you know, the manipulator prototype is the 7-DoF manipulator in Figure 1. For the sake of simplicity and logical reasoning, the authors used the 4-DoF arm to determine the objective functions. Since reducing the problem from 7-DoF to 4-DoF is an important simplification in the approach to determining the design indices, introducing the Denavit-Hartenberg parameters and Jacobian matrix of the 7-DoF manipulator are vital to let the readers keep track of the problem-solving approach. Hence we think that it is better to keep the sections related to the 7-DoF manipulator entirely, otherwise, that may be confusing for the reader.

Comment 2:

There are some notations that can create confusion.

Response to comment 2:

The notations were changed in the revised version of the paper.

Comment 3:

In equations 7-10 which is the meaning of “m”. The notation was used in row 329 as a Jacobian matrix dimension.

Response to comment 3:

The m notation represents the dimension of the Cartesian space  (the workspace) of the robot in both line 329 and equations 7 to 10. The definition is added in the revised version of the paper.

Comment 4:

In equation 10 what is the meaning of the parameter σi?

Response to comment 4:

In equation 10 the parameter is the singular value of the Jacobian matrix of the robotic manipulator. This definition is provided in the revised version of the paper.

Comment 5:

In equations 13 and 14 what is the meaning of parameters A and B?

Response to comment 5:

The globalization method in equations 12 to 14 was proposed in the reference number 15 in which no meaning is provided for A and it is just used to calculate the global version of the index. However, the denominator B is the volume of the robot’s workspace. The definition of B is provided in the revised version of our paper.

Comment 6:

What are the mechanical implications when the real manipulator must work as in the simulation (attached to the wall)?

Response to comment 6:

In fact, the base of the manipulators must be attached to the walls of a platform that straddles the vine row. The platform moves forward as the robots are finished with a vine. The platform is not proposed in this paper but the walls are represented in Figures 9 and 11. However, you can find such a platform in references number 3 and 4. Therefore the word “wall” does not invoke a static wall but rather it is an integral part of a platform. In order to avoid misunderstandings, the concept is clarified in the revised version of the paper.

Reviewer 2 Report

This paper can be published in the present form

Author Response

Dear Reviewer

Sincerest thanks for your response and comments on our manuscript. We have modified the paper in response to the extensive and insightful comments. We have worked on clarification to fully address them. We have stressed the main goal of this paper as the technical presentation of treatment in the abstract and conclusion. The new version is reviewed and edited extensively. 

Reviewer 3 Report

The Paper, aimed to serial mechanism, has been written and structured well gathering the important elements of related subject. However, updating the manuscript as per comments given below may enhance its quality.

1)     The novelty of presented research should be better explained in the introduction.

2)     It is recommended to update literature review referring more recent literature related to the presented subject. 

3)     The English language may be revisited as there are inconsistencies in the singular-plural within the same sentence.

4)     Before proceeding to describe your chosen model and actions (immediately after the Introduction section), it is recommended to describe your scientific hypothesis, concepts and the relevant reasoning for choosing the particular modelling approach. This should be accompanied by an overall description of the followed procedure. A block diagram of the procedure would also be very useful.

5)     The measurement uncertainty may be provided for better understanding of results.

Author Response

Dear Rivewer

Greetings,

Sincerest thanks for your response and comments on our manuscript. We have modified the paper in response to the extensive and insightful comments. We have worked on clarification to fully address the comments. We have stressed the main goal of this paper as the technical presentation of treatment in the abstract and conclusion. 

Comment 1:

The novelty of the presented research should be better explained in the introduction.

Response to comment 1:

We have tried to explain the novelty more effectively in the last paragraph of the introduction in the revised version of the paper.

Comment 2:

 It is recommended to update literature review referring more recent literature related to the presented subject. 

Response to comment 2:

The subject of pruning robots is a very novel field of study, therefore naturally there are not a lot of papers concerning this subject. Hence the authors have tried to perform a proper and precise search for relevant papers and references. Both the authors searched the scientific databases for recently published papers again, which resulted in finding a paper published on arxiv.org on the 14th of June 2022. This new paper is cited as reference number 10 in the revised version of our paper.

Comment 3:

The English language may be revisited as there are inconsistencies in the singular plural within the same sentence.

Response to comment 3:

The new version is reviewed and edited extensively

Comment 4:

Before proceeding to describe your chosen model and actions (immediately after the Introduction section), it is recommended to describe your scientific hypothesis, concepts, and the relevant reasoning for choosing the particular modelling approach. This should be accompanied by an overall description of the followed procedure. A block diagram of the procedure would also be very useful.

Response to comment 4:

We should point out that there was a numbering error in section 2 of the first version of the paper which might have led to some misunderstandings. We have corrected it in the revised version. In addition, section 2 introduces a proper systematic procedure for the kinematic task-based design of any agricultural robotic manipulator including pruning robots, therefore we think that specific points related to the pruning robot design should not be explained in this section. All the hypotheses, concepts, and reasoning related to any design stage of the pruning robot are stated in its relevant section, furthermore restating them results in redundancy in writing (stating the same thing over and over). However as you have stated in this comment, we did not mention our reason for choosing the modeling approach “B” between the two choices. Therefore we have explained our reasoning in the first paragraph of section 4 in the revised version of the paper. Moreover, since in the revised version the numbering of design procedure stages in section 2 is corrected, the design procedure is plainer and there might not be vital to add a block diagram.

Comment 5:

The measurement uncertainty may be provided for better understanding of results.

Response to comment 5:

The measurement of uncertainty is the main concept of the following manuscript, and we are going to use the direct estimation of uncertainty from the final simulation result as well as by going through the usual analytical method of propagation of uncertainty

Reviewer 4 Report

The paper solves a very important and interesting problem and effectiveness is described in detail through the design optimization problem. It is very well written and author has followed well established methodology to conduct the study and answer the well formulated research questions. Results are illustrated in a very nice way using graphs and the result are discussed in detail.

Author Response

Dear Reviewer

Greetings,

Sincerest thanks for your response and comments.